# 🦌 DeeR-VLA: Dynamic Inference of Multimodal Large Language Models for Efficient Robot Execution

**Yang Yue**[* 1]     **Yulin Wang**[* 1]     **Bingyi Kang**[2]     **Yizeng Han**[1]     **Shenzhi Wang**[1]
**Shiji Song**[1]     **Jiashi Feng**[2]     **Gao Huang**[† 1]

[1] Department of Automation, BNRist, Tsinghua University     [2] ByteDance
{le-y22, wang-yl19}@mails.tsinghua.edu.cn  {gaohuang}@tsinghua.edu.cn

## Abstract

Multimodal Large Language Models (MLLMs) have demonstrated remarkable comprehension and reasoning capabilities with complex language and visual data. These advances have spurred the vision of establishing a generalist robotic MLLM proficient in understanding complex human instructions and accomplishing various embodied tasks. However, developing MLLMs for real-world robots is challenging due to the typically limited computation and memory capacities available on robotic platforms. In contrast, the inference of MLLMs involves storing billions of parameters and performing tremendous computation, imposing significant hardware demands. In our paper, we seek to address this challenge by leveraging an intriguing observation: relatively easier situations make up the bulk of the procedure of controlling robots to fulfill diverse tasks, and they generally require far smaller models to obtain the correct robotic actions. Motivated by this observation, we propose a **D**ynamic **E**arly-**E**xit Framework for **R**obotic Vision-Language-Action Model (DeeR-VLA, or simply DeeR) that automatically adjusts the size of the activated MLLM based on each situation at hand. The approach leverages a multi-exit architecture in MLLMs, which allows the model to terminate processing once a proper size of the model has been activated for a specific situation, thus avoiding further redundant computation. Additionally, we develop novel algorithms that establish early-termination criteria for DeeR, conditioned on predefined demands such as average computational cost (*i.e.*, power consumption), as well as peak computational consumption (*i.e.*, latency) and GPU memory usage. These enhancements ensure that DeeR operates efficiently under varying resource constraints while maintaining competitive performance. Moreover, we design a tailored training method for integrating temporal information on top of such multi-exit architectures to predict actions reasonably. On the CALVIN robot manipulation benchmark, DeeR demonstrates significant reductions in computational costs of LLM by 5.2-6.5x and GPU memory of LLM by 2-6x without compromising performance. Code and checkpoints are available at https://github.com/yueyang130/DeeR-VLA.

## 1   Introduction

The recent astonishing progress in multimodal large language models (MLLMs) has unveiled their remarkable potential of extracting, aligning, and integrating the representations from complicated language and visual data [1, 2, 3, 4]. These advances have spurred the vision of a generalist robot, *i.e.*, an embodied agent equipped with vision-language comprehension and problem-solving capabilities,

---

*Equal contribution. †Corresponding author.

38th Conference on Neural Information Processing Systems (NeurIPS 2024).

proficient in interacting with humans and the physical world to flexibly execute complex manipulation tasks [5, 6]. An encouraging preliminary work, RT-2 [7, 8], has demonstrated the feasibility of adopting MLLMs to control robots in end-to-end. This not only yields performant robotic policies, but also exhibits some emergent abilities obtained from large models, such as understanding novel commands, generalizing to objects never seen before, and reasoning.

Despite these favorable findings, the high demands of MLLMs on hardware are usually an important bottleneck that inhibits the establishment of generalist robots with advanced MLLMs. Typically, robotic applications are based on resource-hungry platforms with limited computational capability, memory space, and battery capacity, yet usually necessitate acting in real-time and performing low-latency interactions with humans or the physical environments. However, every time MLLMs are activated to obtain a robotic action involves utilizing billions of parameters to accomplish a computationally intensive inference process. Such inefficiencies may lead to considerable GPU memory requirements, tremendous power consumption, as well as nontrivial time delays in controlling robots. These weaknesses make it challenging to deploy MLLMs on real embodied robotic systems.

To alleviate this problem, we propose an approach based on dynamic neural networks. Our work is inspired by an intriguing observation: in the procedure of controlling a robot to fulfill various tasks, relatively 'easier' circumstances make up the bulk of all the situations confronted by the robot. When encountered with these 'easier' situations, an embodied agent can actually acquire proper robotic actions with a much smaller model compared to the full MLLMs. Or more precisely, only the remaining small number of 'more difficult' circumstances necessitate the full capacity of large MLLMs. This phenomenon can be demonstrated using the representative example in Table 1, where we train RoboFlamingo [9] with varying model sizes, and report the FLOPs and task successful rate in the Calvin Long-Horizon Multi-Task Language Control (LH-MTLC) challenge [10]. Adopting the officially recommended 24-layer Flamingo only correctly finishes 3.2% (78.9% v.s. 75.7%) more tasks compared to using 6 layers, but it increases the computational cost by 4x. In other words, computational resources are wasted on activating larger models in many easy circumstances for which smaller models are sufficient.

Motivated by this observation, we propose a *Dynamic Early-Exit for Robotic MLLM* (DeeR) framework, seeking to automatically configure the size of MLLMs conditioned on each situation confronted by an embodied agent. Specifically, we introduce a MLLM architecture featuring multiple intermediate exits, with which a correct robotic action can be immediately obtained once a proper size of the model has been activated, eliminating further redundant computation. Additionally, we develop novel algorithms that are able to establish early-termination criteria for DeeR conditioned on arbitrarily specified demands of average computational cost (*i.e.*, power consumption), and peak computational cost (*i.e.*, latency) or GPU memory overhead. At inference,

Table 1: Computation cost v.s. task successful rate[1](RoboFlamingo++) on CALVIN LH-MTLC chanllenge D→D. Notably, we mainly focus on the core component, LLM, of the MLLM, which comprises the majority of parameters. We vary the size of the LLM to examine its impact. For a focused comparison, we report the FLOPs (and GPU memory usage) of the LLM in our paper, unless otherwise specified.

| # LLM layers | 24 | 12 | 6 |
|---|---|---|---|
| GFLOPs/action (LLM) | 31.2 | 15.6 | 7.8 |
| Task success rate % | 78.9 | 78.0 | 75.7 |

DeeR can adaptively activate smaller models for less complex situations and larger models for more challenging cases. Consequently, computation is unevenly allocated among situations, yielding a considerable improvement in efficiency. Besides, the computational cost of DeeR can be adjusted online by simply modifying the termination criterion on top of a fixed main model, making it appealing in flexibility. Moreover, we design a tailored training method for DeeR, enabling integrating temporal information on top of such multi-exit architectures to control robots reasonably.

The performance of DeeR is evaluated on 3 CALVIN LH-MTLC challenges with RoboFlamingo [9]. Extensive robot experiments show that DeeR reduces the LLM computational cost by 5.2-6.5x without sacrificing performance. Surprisingly, even when considering GPU memory limitations in the termination criterion, DeeR remains competitive with other SOTA methods while only utilizing 2GB memory for the activated LLM. Consequently, DeeR demonstrates the potential to enable a wider range of users to operate their own robots equipped with MLLMs on resource-limited platforms.

---

[1]Average successful rate over all subtasks in the long-horizon chains.

## 2 Related Works

**LLM/MLLM for language-conditioned robot control.** A range of studies have explored the use of natural language to instruct robots in performing tasks [11, 12, 13, 14, 15, 16]. Methods such as SayCan and PaLM-E [17, 18, 19] utilize LLMs as high-level planners to translate commands into individual primitives that are then executed by low-level controllers. However, these controllers are usually domain-specific small models and lack the semantic understanding and reasoning capabilities that LLMs/MLLMs possess. To fully leverage LLM's astonishing capabilities, RT-2 [7, 9] proposes an end-to-end MLLM that directly generates low-level robotic actions via co-finetuning on robotic data and image-language data. It exhibits some emergent abilities obtained from large MLLMs, such as generalizing to instructions and objects never seen before, and reasoning. Further, RoboFlamingo [9] proposes to adapt existing MLLMs to a low-level robotic policy through straightforward fine-tuning on robotics data. While representative projects like RT-2 and RoboFlamingo have showcased the promising potential in enabling generalist robots, the use of MLLMs for such low-level control is computationally intensive. This is because each robotic action requires processing through all layers of an MLLM, whose inefficiencies often yield significant bottlenecks in practical robotic applications.

**Efficient LLM.** Considerable strides have been made to improve the inference efficiency of LLMs [20, 21, 22]. Research in this domain typically falls into three categories: efficient structural design[23, 24, 25, 26, 27, 28, 29], model compression [30, 31, 32, 33, 34, 35, 36], and dynamic networks [37, 38, 39, 40]. Our research focuses on the third category, dynamic networks, which optimize computational resources based on input data to reduce unnecessary computation. A key strategy within this category is *early exiting*, discussed further below.

**Early exiting** is an innovative method for dynamically halting forward propagation at a certain layer based on intermediate predictions. This technique has been well explored in both computer vision [41, 42, 43, 44, 45, 46], language processing [47, 48, 49, 50, 51, 52, 53], and multimodality [54, 55]. A challenge in implementing early-exiting models is devising an appropriate metric to determine when to issue an intermediate prediction. Traditionally, in tasks such as image classification, metrics such as Softmax confidence [43] or entropy [49] are utilized. Alternative approaches include training learning-based exit strategies with pseudo labels [56, 48, 57, 58]. Recent advancements [38, 39] have expanded early exiting to encompass the next-token prediction of LLMs focused on, treating it as a classification task. Diverging from these methods, our work adapts an MLLM to generate action outputs for sequential decision-making. We introduce a novel dynamic paradigm that integrates temporal information to predict action. Further, we devise a novel early-exiting metric based on action consistency, necessary because typical metrics like confidence and entropy are infeasible without direct Softmax outputs. Lastly, we develop an algorithm to derive termination criteria through online environmental interaction—a strategy not explored in prior early-exiting research in vision or NLP.

## 3 Dynamic Early-Exit for Robotic MLLM

The strong task instruction understanding and visual grounding capabilities of MLLMs [5, 4] have exhibited great promise for language-instructed multitask robotic manipulation [7, 8, 9]. However, existing works tend to be computationally intensive since the actions of the robot are obtained by inferring all layers of an MLLM. At each timestep, this process may activate billions of parameters, necessitating substantial computation and memory, and yielding a significant latency and power consumption. These inefficiencies are usually important bottlenecks for practical robotic applications.

**Overview.** We seek to address this challenge by leveraging an intriguing observation: relatively 'easier' situations make up the bulk of the procedure of controlling robots to fulfill various tasks, and they generally require far smaller models to obtain the correct robotic actions (as shown in Table 1). Inspired by this phenomenon, we propose *Dynamic Early-Exit for Robotic MLLM* (DeeR), aiming to improve the computational efficiency of the robotic MLLM systems by dynamically adopting a proper size of MLLM for each situation. In specific, we first develop a novel MLLM architecture with multiple intermediate exits (Section 3.1). Consequently, given an input, one can immediately acquire a proper robotic action once a sufficient number of model parameters have been activated, avoiding further redundant computation. Then, Section 3.2 establishes early-termination criteria for DeeR conditioned on arbitrarily specified demands of average computational cost, and peak computational cost or GPU memory overhead. Finally, Section 3.3 proposes a tailored training algorithm for our model, demonstrating how to integrate temporal information on top of this dynamic network and reasonably predict robotic actions.

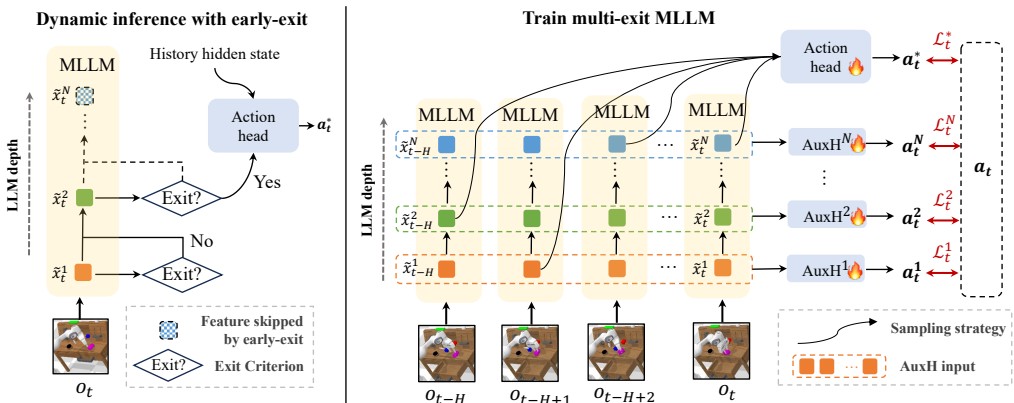

Figure 1: **Left**: Dynamic inference of DeeR. For inference, we adaptively activate an appropriate size of MLLM based on an exit criterion $c$, which accounts for the *current situation* (including task instruction $l$ and observation $o_t$) and *predefined computational and GPU memory budgets*. The language instruction and gripper camera image, not shown in this figure, are also inputs to the MLLM. An action is then obtained using the intermediate feature $\tilde{x}_t^{c(t)}$ and historical information. **Right**: Training of DeeR. We randomly sample features from all exits during training. This strategy helps minimize the discrepancy between training and dynamic inference. Moreover, we employ several auxiliary action heads (AuxH) to better optimize the MLLM.

## 3.1 Multi-exit Architecture for Robot

We first introduce an MLLM architecture featuring multiple intermediate exits, enabling the dynamic adaptation of the MLLM's size to suit the varying situations encountered by robots.

**Basic architecture.** Tasked with a language instruction $l$, a robot receives an observation $o_t$ from sensors (*e.g.*, RGB image from the camera in our paper) at timestep $t$ and predicts an action $a_t^*$ to execute. To correctly predict the action, the robot should not only sufficiently understand the language instructions, but also extract task-relevant information from the images [59]. Built upon an existing work [9], we achieve this by employing a pretrained MLLM, *i.e.*, Flamingo [60, 61], to process and integrate both vision and language inputs, thus obtaining fused multimodal features for decision-making.

Our basic MLLM mainly consists of a vision encoder $E_I$ and a LLM. The vision encoder $E_I$ comprises a

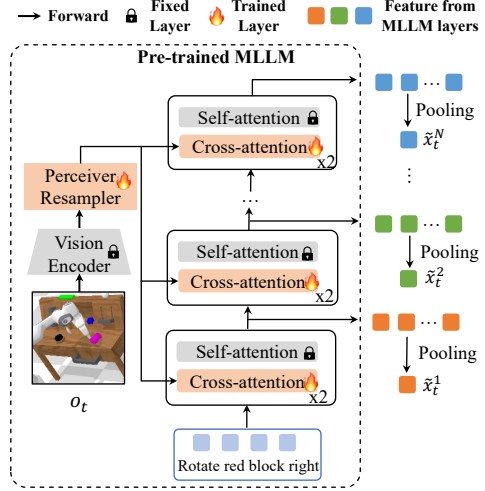

Figure 2: Multi-exit MLLM architecture for robot.

Vision Transformer (ViT) [62] paired with a Perceiver Resampler [60], which encodes an input image $o_t$ into a sequence of informative tokens. For multimodal fusion, an LLM is established on top of the visual representations generated by the vision encoder $E_I$. More specifically, we interleave the self-attention blocks of a pretrained, frozen text-only LLM with newly introduced, learnable cross-attention blocks that cross-attend to the visual tokens. This configuration allows the original MLLM to function as an effective multimodal feature extractor $F_\theta$, formalized as follows:

$$x_t = F_\theta(l, E_I(o_t)), \tag{1}$$

where $l$ denotes the input language instruction tokens with a length $L$, and the output $x_t = (x_{t,1}, x_{t,2}, \ldots, x_{t,L})$ represents the hidden state sequence from the last layer of our MLLM at timestep $t$. Notably, despite the effectiveness of LLMs in multimodal feature integration, their reliance on billions of parameters results in substantial computational costs and memory usage.

**Visual language model with intermediate exits.** We dynamically adapt the size of the LLM to the specific requirements of each situation encountered by a robot by introducing a model with intermediate exits. Specifically, we divide the LLM layers into $N$ consecutive groups, noted as $F_\theta^1, F_\theta^2, \ldots, F_\theta^N$. Each group $F_\theta^i$ outputs an intermediate hidden state sequence $x_t^i = (x_{t,1}^i, x_{t,2}^i, \ldots, x_{t,L}^i)$. When computation terminates at an intermediate exit $i$, we apply a max-pooling operator $P$ to aggregate the information across the token dimension, resulting in a compact representation $\tilde{x}_t^i = P(x_{t,1}^i, x_{t,2}^i, \ldots, x_{t,L}^i)$

that effectively summarizes the image $o_t$ and instruction $l$. This representation serves as the input for the subsequent action prediction module. With such a multi-exit MLLM architecture, we can obtain a series of informative representations $\tilde{x}_t^1, \tilde{x}_t^2, ..., \tilde{x}_t^N$ at varying scales of LLM processing. This allows us to dynamically select the most suitable LLM size conditioned on the situation complexity without activating parameters beyond the chosen exit point. Our multi-exit MLLM is illustrated in Figure 2.

**Predicting robotic actions with an action head.** After the LLM processes to an appropriate level, the output $\tilde{x}_t^i$ from the $i$-th intermediate exit is transformed into low-level actions by a lightweight action head. In this paper, we consider a 7 DoF end-effector action as a representative example of low-level actions, where the first six continuous dimensions specify the position and orientation of the end-effector, and the seventh discrete value indicates whether the gripper is open or closed. Notably, given that the decision-making environment is typically characterized as a Partially Observable Markov Decision Process (POMDP) [63], optimal decisions rely not only on the current observation $o_t$ but also on historical observations. Thus, we employ a sequence model as the action head $\pi_\theta$ to aggregate information across a history window of size $H$. Without loss of generality, this paper considers a lightweight LSTM [64] as an example. On the top of LSTM are two distinct MLP modules: one dedicated to predicting the pose of the end-effector, and the other to predict the discrete gripper status. The lightweight action head $\pi_\theta$ computes actions efficiently with minimal computational overhead.

**Early-terminated inference of robotic actions.** Equipped with the action head, we assume that a criterion $c$ is defined to determine the optimal point for the conditional exiting from an appropriately sized LLM at the current timestep $t$ (we will discuss the details of criteria in Section 3.2). The index of the selected exit, denoted as $c(t)$, ranges from 1 to $N$. Consequently, we utilize the feature $\tilde{x}_t^{c(t)}$ from the $c(t)$-th LLM block to compute the predicted action $a_t^*$ for the current timestep as follows:

$$a_t^*, h_t = \pi_\theta(\tilde{x}_t^{c(t)}, h_{t-1}), \qquad (2)$$

where $h_t$ represents the LSTM's hidden state, with $h_0$ initially set to a zero vector. The predicted action $a_t^*$ is composed of pose action and gripper action.

## 3.2 Adaptive Inference

This section demonstrates how DeeR efficiently executes robot tasks by adaptively activating a proper size of the MLLM under predefined computation and GPU memory budgets. Specifically, we first discuss the termination criterion utilized by DeeR, designed to activate smaller models for less complex scenarios and larger models for more challenging conditions. Next, we explore our approach to devising an effective resource allocation strategy that addresses limitations in computation and GPU memory. The inference process of DeeR is illustrated in Figure 1.

**Termination criterion.** As mentioned in related works, many previous works utilize confidence-based criteria for determining when to terminate, typically involving metrics such as the maximum element or entropy of the SoftMax output [37, 43, 65, 66, 49]. In our case, where the goal is action prediction and SoftMax output is not readily available, we adopt a different approach by leveraging the consistency of action predictions from adjacent intermediate features as our criterion. The underlying intuition is that if the action predictions from two differently sized MLLMs remain consistent, it suggests that the computational model may have reached saturation, and further processing is unlikely to yield any further improvements. For a given timestep $t$, we identify the smallest $i$ within the range $[1, 2, ..., N]$ that satisfies the following action consistency condition as termination exit:

$$\|\pi_\theta(\tilde{x}_t^i, h_{t-1}) - \pi_\theta(\tilde{x}_t^{i-1}, h_{t-1})\|_2 < \eta_i, \qquad (3)$$

where we disregard the hidden state outputs of $\pi_\theta$ and focus solely on comparing the $L2$ norm of the difference in predicted actions against a predefined threshold $\eta_i$. We always adopt infinity as $\eta_N$ to ensure all samples can exit. For $i = 1$, we use the input features to the LLM layer as $x_t^{i-1}$.

**Budgeted task execution.** Given the predefined constraints of computation and memory budgets, it can be challenging to manually set optimal threshold values $\{\eta_1, \eta_2, ...\}$ to ensure that the robotic MLLM policy achieves peak performance while adhering to budget limitations. In contrast, we propose to determine these values by formulating an optimization problem. We operate under a *Budgeted Task Execution Setting*, where DeeR is required to perform a set of tasks $\mathcal{T}$ within a specified total computational budget $B > 0$. To ensure that each action is delivered within an acceptable waiting time, we impose constraints on peak computation where $G > 0$. Additionally, we limit GPU memory usage to $M > 0$ to accommodate scenarios where users may not have access to high-memory GPUs. Let $\text{Scc}(\mathcal{T}, \{\eta_1, \eta_2, ...\})$ represent the success rate of tasks in $\mathcal{T}$, and let

FLOPs($\mathcal{T}, \{\eta_1, \eta_2, \ldots\}$) denote the computational cost for executing these tasks under specified constraints. Furthermore, MFLOPs($\mathcal{T}, \{\eta_1, \eta_2, \ldots\}$) denotes the peak FLOPs across all timesteps, and Mem($\mathcal{T}, \{\eta_1, \eta_2, \ldots\}$) indicates GPU memory used during task execution. We seek the optimal thresholds by addressing the following optimization problem:

$$\max_{\eta_1, \eta_2, \ldots} \text{Scc}(\mathcal{T}, \{\eta_1, \eta_2, \ldots\}), \tag{4}$$

subject to

$$\text{FLOPs}(\mathcal{T}, \{\eta_1, \eta_2, \ldots\}) < B, \qquad \text{(average FLOPs constraint)}$$
$$\text{MFLOPs}(\mathcal{T}, \{\eta_1, \eta_2, \ldots\}) < G, \qquad \text{(peak FLOPs constraint)}$$
$$\text{Mem}(\mathcal{T}, \{\eta_1, \eta_2, \ldots\}) < M. \qquad \text{(GPU memory constraint)}$$

Due to the non-differentiability of the success rate function $\text{Scc}(\cdot, \cdot)$, we may leverage heuristic algorithms to solve for the thresholds that maximize success within the computational constraints. We discuss strategies for determining optimal thresholds under two conditions: one where we only have access to a demonstration dataset, and another where real environment interaction is permitted.

**Solving problem (4) using a demonstration dataset.** We denote by $0 < q \leq 1$ the probability that a sample reaching an exit point will meet the termination criterion and thus exit at that point. When accessing only a demonstration dataset, we assume $q$ is constant across all layers [43]. This suggests that the proportion of samples exiting at exit $i$ can be represented as $q_i = zq^i$, where $z$ is a normalizing constant ensuring that $\sum_{i=1}^{n} q_i = 1$. Here, $n \leq N$ denotes the maximum allowable exit index where the corresponding activated LLM meets the constraints of peak GFLOPs and GPU memory. The proportions of samples at the exits whose index are greater than $n$ is set to zero. At testing time, we must adhere to the computational budget constraint:

$$|\mathcal{T}| \overline{L} \sum_{i=1}^{n} q_i C_i \leq B, \tag{5}$$

where $|\mathcal{T}|$ is the number of tasks to perform, $\overline{L}$ represents the average length of tasks as derived from the dataset statistics and $C_i$ is the computational cost when the LLM inference terminates at $i$-th exit. Equation (5) allows us to solve for $q$ and determine $q_i$. Using these target proportions $q_i$ for each exit, we then determine the threshold values $\eta_i$ on the dataset to ensure that approximately $q_i$ proportion of timesteps exits at the $i$-th exit [43].

**Solving with online interactions.** If the interaction with a real environment is feasible, we can utilize online learning algorithms that iteratively adjust thresholds based on feedback regarding success rates. To solve Equation (4) under budget constraints, we implement Bayesian Optimization [67]. We construct the objective function for Bayesian Optimization to maximize as follows:

$$f_{\text{obj}} = \text{Scc}(\mathcal{T}, \{\eta_1, \eta_2, \ldots\}) - P, \tag{6}$$

where $P = 0$ if budget constraints are satisfied otherwise a significant penalty term. This online paradigm allows us to determine thresholds without assuming an exponential distribution, enabling the acquisition of more effective thresholds through real-time feedback. Here, we employ Bayesian Optimization as a representative example of an online solving approach. Future work could extend to other online algorithms such as multi-armed bandit or reinforcement learning [68].

### 3.3 Training Algorithm

Notably, it is nontrivial to train our dynamic robotic MLLM properly. Specifically, dynamic adjustment of the network architecture leads to a *discrepancy* between training and inference. During inference, we use a deterministic criterion to select a proper intermediate feature at each timestep. Nevertheless, during training, we lack a well-defined termination criterion and remain unaware of the distribution of features across the exits. To enable our model to learn to integrate temporal information effectively, we propose a tailored training algorithm, as introduced in the following.

**Learning with an arbitrary size of models.** To reduce the aforementioned discrepancy, we propose a simple yet effective random sampling strategy during training. As depicted by the "winding" curves on the right side of Figure 1, our approach involves sampling an exit index from 1 to $N$ at each timestep. We implement two types of sampling strategies. *The first strategy*, denoted as $s_1$, is to uniformly sample an exit index from 1 to $N$ at each step. This ensures that features from all possible exits are effectively captured in the action head during training. It simulates scenarios where the action head might encounter features from all exits within a given time window, thus accommodating an

Table 2: Comparison with baselines. GR-1 uses extra proprioceptive information as input. Note that some baselines mainly focus on one or two settings, and we present results following their original papers. We report the performance of our method at the last epoch. The value in parentheses indicates the LLM FLOPs required to achieve the reported score. The success rates for the 1st to 5th subtasks are in Appendix B.1.

| Method | Input | Data | Foundation model | Avg. successful len (*LLM GFLOPs*) | | |
| | | | | D→D | ABCD→D | ABC→D |
| --- | --- | --- | --- | --- | --- | --- |
| GR-1 [69] (*ICLR'24*) | RGB+Proprio | LANG | Video-pretrained Transformer | - | 4.21 | 3.06 |
| HULC [13] (*RA-L'22*) | RGB | ALL | ✗ | 2.64 | 3.06 | 0.67 |
| RT-1 [15] (*RSS'23*) | RGB | LANG | ✗ | - | 2.45 | 0.9 |
| SPIL [70] (*ICML'24*) | RGB | ALL | ✗ | 2.67 | - | 1.71 |
| SuSIE [71] (*ICLR'24*) | RGB | ALL | InstructPix2Pix [72] | - | - | 2.69 |
| RoboFlamingo (*ICLR'24*) | RGB | LANG | OpenFlamingo 3B | 2.46 (*31.2*) | 4.08 (*31.2*) | 2.47 (*31.2*) |
| RoboFlamingo++ | RGB | LANG | OpenFlamingo 3B | 2.71 (*31.2*) | 4.07 (*31.2*) | 2.59 (*31.2*) |
| DeeR (ours) | RGB | LANG | OpenFlamingo 3B | **2.83** (*8.6*) | **4.13** (*10.0*) | **2.82** (*12.5*) |
| DeeR w. online (ours) | RGB | LANG | OpenFlamingo 3B | **2.92** (*8.5*) | **4.13** (*9.7*) | **2.90** (*9.5*) |

arbitrary inference pattern and reducing the training-inference discrepancy. Moreover, we observe that in practice, the dynamic model often terminates at the same exit for multiple consecutive timesteps, as the neighboring observations tend to be quite similar. The model then switches to another exit for a sequence of subsequent timesteps. To better emulate this pattern during training, we adopt *a second sampling strategy* denoted as $s_2$. Specifically, we split the time window $o_{t:t+H-1}$ into two consecutive segments $o_{t:t+i}$ and $o_{t+i+1:t+H-1}$, with $i$ chosen randomly. In each segment, a single uniformly sampled index is assigned and shared across all timesteps.

On top of these two sampling strategies, we can define our training loss function. We sample from a robot demonstration dataset a language instruction $l$ and a clip of observation-actions $\{o_t, a_t, o_{t+1}, a_{t+1}, \ldots, o_{t+H-1}, a_{t+H-1}\}$. For each sampling strategy $s \in \{s_1, s_2\}$, we use Equation (1) and Equation (2) to predict each action $a_{t+i}^{*,s}$ for $s = s_1, s_2$ and $i = 0, 1, \ldots, H-1$, where $c(t)$ in Equation (2) is replaced by the sampling strategy $s$. For each pair of the predicted action $a^*$ and the actual action $a$, we define a single-action loss function $\mathcal{L}(a^*, a)$ that incorporates both mean squared error (MSE) for pose prediction and cross-entropy loss for gripper status prediction with a coefficient $\lambda$ to balance the two terms [9]. The total loss for a sequence is then the sum of the losses over timesteps:

$$\mathcal{L}^* = \sum_{s \in \{s_1, s_2\}} \sum_{i=0}^{H-1} \mathcal{L}(a_{t+i}^{*,s}, a_{t+i}). \qquad (7)$$

**Auxiliary losses.** The intermediate features from the original MLLM, intended as input for subsequent layers, may not be optimal for output prediction. To ensure that each activated size of the MLLM in our framework produces features suitable for predicting actions, we introduce auxiliary losses. Specifically, we attach $N$ auxiliary action heads (denoted as UAH in Figure 1) at the exits. The $j$-th auxiliary head processes temporal features from the $j$-th exit and predicts the action $a_t^j$. We jointly train the auxiliary heads and the MLLM using the loss function:

$$\mathcal{L}_{\text{aux}} = \sum_{j=1}^{N} \sum_{i=0}^{H-1} \mathcal{L}(a_{t+i}^j, a_{t+i}). \qquad (8)$$

These auxiliary heads are employed only during training and are not used for inference.

**Total Loss.** The full training pipeline is depicted in Figure 1. We fine-tune only the parameters of the perceiver sampler and cross-attention layers in the MLLM, with the randomly initialized action head $\pi_\theta$ and auxiliary action heads. The visual encoder and the other LLM components, remain frozen. The total loss for the training process is expressed as $\mathcal{L}_{\text{total}} = \mathcal{L}^* + \mathcal{L}_{\text{aux}}$.

## 4 Experiments

**Setup.** In this section, we conduct experiments to validate the effectiveness of DeeR as an efficient robot policy. Specifically, we build DeeR upon the RoboFlamingo++ codebase. We preserve hyper-parameters from RoboFlamingo++ for fair comparison, except for the number of LLM layers and our proposed dynamic early-exit paradigm. we compare DeeR in terms of budget *v.s.* performance with similarly sized RoboFlamingo++ models [9] and other SOTA baselines. We provide implementation details in Appendix A.

**Measures of efficiency.** In modern foundation models, the LLM typically plays a pivotal role within an MLLM in terms of reasoning and problem-solving tasks, and it usually contains the majority of the

model's parameters [4, 61]. Our work mainly focuses on improving the efficiency of LLMs within a robotic context. To facilitate a focused comparison, in our experiments, we report the number of floating point operations (FLOPs) and GPU memory usage for LLM's inference.

**Benchmark.** We utilize the CALVIN Long-Horizon Multi-Task Language Control benchmark (LH-MTLC) [10] as the testbed to test our learned multi-task, language-conditioned policy. In the CALVIN, the objective is for the agent to successfully complete task sequences, each with five subtasks described in natural language. Following previous works [10, 12, 13, 9], model performance is evaluated based on the average successful length (0 to 5) across 1000 task sequences.

**Datasets.** The CALVIN dataset [10] is divided into four environmental splits, labeled A through D, each characterized by unique backgrounds and object configurations. Each split contains over 2 million robot manipulation trajectories (denoted as '*ALL*'). Of these, only about 1%, approximately 24 thousand trajectories, are annotated with language instructions (denoted as '*LANG*'). For training DeeR, we exclusively utilize the 'LANG' data. In our study, we evaluate models across three settings to thoroughly assess their imitation and generalization capabilities: 1) D→D: Train and evaluate in a single environment, 2) ABC→D: Zero-Shot Multi-Environment, 3) ABCD→D: Multi-Environment.

**Baselines.** For a comprehensive comparison, we consider various baselines. We include HULC [13] and SPIL [70] as representatives of approaches reliant on hierarchical planning and skill priors [12, 14, 73]. Additionally, we evaluate models using pretrained or foundation models, such as RT-1 [15], SuSIE [71], GR-1 [69], and RoboFlamingo [9]. RoboFlamingo++ is our reproduced RoboFlamingo.

## 4.1 Main Results

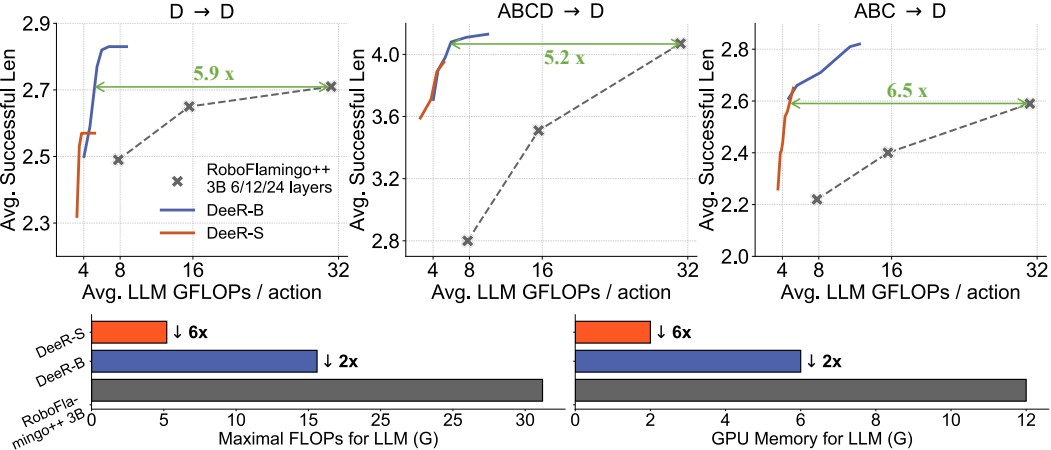

Figure 3: Results atop OpenFlamingo 3B. **Upper**: Avg. successful len v.s. avg. LLM GFLOPs. **Bottom**: Peak GLOPs and GPU memory for LLM. Different colors indicate different peak FLOPs and GPU memory budgets, denoted as DeeR-S and DeeR-B (they share a fixed model). DeeR preserve all the architecture and hyperparameters from RoboFlamingo++ for fair comparisons, except for our dynamic early-exit paradigm.

**Results on Flamingo 3B** are presented in Figure 3. We train just a single model in each CALVIN setting. Given the predefined total computational budget $B$, the maximum FLOPs $G$, and the GPU memory $M$, we adhere to these budgets by adjusting the termination thresholds, which are determined by solving Equation (4) with the CALVIN dataset. Then we assess the average successful length of DeeR under different thresholds to plot the curves. It can be observed that DeeR consistently reduces the computational cost of the LLMs across all settings. For instance, in the setting D→D, DeeR achieves an average successful length 2.71 with 5.9x fewer average FLOPs, 2x fewer maximum FLOPs and 2x fewer GPU memory. Surprisingly, DeeR-S achieves relatively high performance with only 2GB memory consumed by LLM, which is affordable to most users. Thus, DeeR demonstrates the potential to enable a broader range of users to operate their own robots effectively with LLMs.

**Comparison with SOTA baselines.** In Table 2, we benchmark the DeeR model against recently SOTA methods in the CALVIN benchmark. Our analysis reveals that DeeR achieves competitive performance compared to the latest SOTA model GR-1 which uses additional proprioceptive information. When compared with traditional imitation learning methods without foundation model, DeeR demonstrates superior performance, particularly in generalization scenarios (ABC→D). Moreover, DeeR slightly outperforms RoboFlamingo while requiring less computation.

**Solve thresholds with online interaction.** When interaction with the environment is feasible, we utilize Bayesian Optimization to solve Equation (4) as we stated in Section 3.2. As shown in Table 2, we discovered that finding thresholds via online interaction is particularly effective in challenging scenarios such as low-data environments (D→D) and generalization to unseen situations (ABC→D).

**Scalability of DeeR.** We developed DeeR on top of OpenFlamingo 9B [61] to evaluate its efficiency when scaling up the foundation model. The results, detailed in Figure 4, indicate that DeeR reduces 1.8-5.7x computation and 2.7x-4.0x peak FLOPs and memory for the same performance.

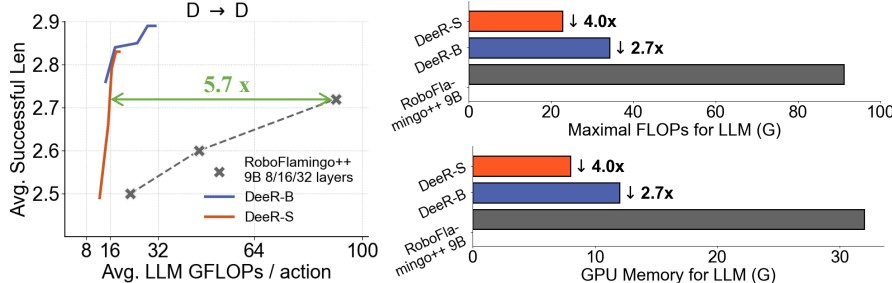

Figure 4: Results on the top of OpenFlamingo 9B. **Left**: Avg. successful len *v.s.* average LLM GFLOPs. **Right**: Maximum GLOPs and GPU memory budget for DeeR-S and DeeR-B. The activated LLM in DeeR-S and DeeR-B consumes 12GB memory, whereas RoboFlamingo 9B requires 32GB.

## 4.2   Ablation Study

**Auxiliary losses.** In this study, we explore the effect of auxiliary losses using the ABCD→D setting as a representative scenario. As shown in Table 3, the model trained without auxiliary losses demonstrates much lower performance. The drop may stem from insufficient optimization for features across exits: The training paradigm without auxiliary losses optimize only one feature from a single exit at each timestep (chosen by sampling strategy) for action prediction.

Table 3: Ablation study of auxiliary losses on ABCD→D.

| GFLOPs | avg. succss len | |
|---|---|---|
| | DeeR | w.o. aux |
| 4.9 | **3.94** | 2.64 |
| 10.0 | **4.13** | 2.71 |

We also observed that in smaller datasets, such as in the D→D setting, omitting auxiliary losses has little to no impact on performance. This may be because smaller datasets are easier to fit, reducing the necessity for auxiliary losses.

**Early-termination criterion.** Based on a fixed DeeR model, we explore various criteria for adaptive inference. We consider using cosine similarity between exit points to determine termination [55]. Specifically, if the similarity value exceeds a threshold, the process is terminated. We introduce another metric that progressively increases the size of the activated LLM as a task progresses, based on the observation that the initial stage of a task generally present simpler scenarios. Our results, detailed in Table 4, demonstrate that our straightforward yet effective action consistency criterion outperforms other criteria across several average computational budgets.

Table 4: Ablation study of exit criteria. Comparing feature similarity, time, and action consistency.

| Settings | GFLOPs | avg. succss len | | |
|---|---|---|---|---|
| | | feat. | time | action |
| D→D | 4.9 | 2.52 | 2.35 | **2.65** |
| | 9.1 | 2.62 | 2.82 | **2.83** |
| ABCD→D | 4.9 | 3.66 | 3.92 | **3.94** |
| | 9.1 | 3.92 | 4.08 | **4.10** |
| ABC→D | 4.9 | 2.29 | 2.46 | **2.62** |
| | 9.1 | 2.45 | 2.71 | **2.75** |

**Real inference efficiency.** We conducted evaluations of real-world operational efficiency. Both RoboFlamingo++ and DeeR were tested on the same Nvidia V100 GPU. As shown in Table 5, DeeR achieved a 68.1% reduction in LLM inference time compared to RoboFlamingo++ (abbreviated as Robo++ in Table 5) when both models achieved the same performance, which aligns with the

Table 5: Comparison of real inference efficiency on the ABCD→D dataset. The average LLM inference time is reported.

| Model | Len | GFLOPs | Time |
|---|---|---|---|
| Robo++ | 4.07 | 31.2 | 55ms |
| DeeR | 4.08 | 6.0 | 17.5ms |

theoretical 80.7% reduction in FLOPs. This evaluation was performed without code optimizations for early-exit implementation. We expect that with further optimizations, DeeR's real-world operational efficiency will improve, potentially aligning even more closely with the predicted FLOPs reduction.

**DeeR with Quantization.** Model compression techniques, such as quantization and pruning, along with efficient structural designs, complement early-exit strategies like DeeR. These methods improve efficiency from different perspectives: quantization reduces memory usage by lowering parameter precision, while early-exit strategies like DeeR reduce computational load by dynamically skipping unnecessary layers. In Table 6, we present quantization as an example to illustrate how DeeR can integrate with these techniques to reduce memory cost.

Table 6: DeeR with quantization on the ABCD→D setting.

| DeeR | Memory | Avg Len |
|---|---|---|
| float32 | 6G | 4.13 |
| float16 | 3G | 4.12 |
| int4 | 1.7G | 3.91 |

### 4.3 Visualization

Figure 5 displays rollouts of DeeR with the termination points. Situations with a higher exit index are considered "harder" by DeeR and thus are allocated more computational resources. One can observe that "hard" situations often involve relatively complex and delicate operations, while "easy" situations typically involve straightforward movements toward target objects. For example, in the task of stacking blocks (1st row), lifting the blue block from the table (1st image) and placing it down on the pink block (images 4 and 5) are allocated more computation, whereas simply moving towards the pink block (images 2 and 3)requires only the smallest LLM to handle. Similar observations occur in the 2nd and 3rd rows, where the stage of moving toward the target object require minimal computation, while pushing the lightbulb switch or moving the sliding door are sophisticated operations that necessitate more LLM processing.

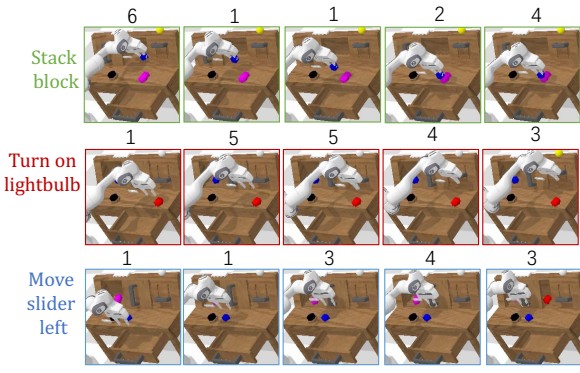

Figure 5: Visualization of DeeR rollouts in the CALVIN environment. Please zoom in to view details. The numbers indicate the termination exit index. Situations with a lower exit index are recognized as 'easier' ones.

## 5 Conclusion and Limitations

In this paper, we introduced the *Dynamic Early-Exit for Robotic MLLM* (DeeR) framework, aiming to dynamically configure the size of MLLMs based on the specific requirements of each situation encountered by a robotic agent. In specific, we proposed a novel MLLM architecture with multiple intermediate exits. Further, we establish early-termination criteria for DeeR based on action consistency and solve thresholds via a dataset or online interaction. Additionally, we crafted a tailored training method that integrates temporal information within this multi-exit framework to enhance robotic control. Extensive robotic experiments demonstrated that DeeR significantly reduces LLM computational costs and GPU memory usage, highlighting its potential to empower a broader range of users to manage their robots on resource-constrained platforms. While our study shows promising results, it has some limitations. We focused on improving LLM efficiency for robotic execution since LLMs account for most of the parameters and GFLOPs. However, the computational cost of the visual encoder is also significant. We expect this limitation to be alleviated as more efficient, lightweight visual encoders are developed. Besides, our experiments were limited to a simulated benchmark. Future work will aim to improve the inference efficiency of the entire MLLM-based robotic system in realistic environments.

## 6 Acknowledgement

The Tsinghua University team is supported in part by the National Key R&D Program of China (2022ZD0114903).

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

# A  Implementation Details

We developed DeeR based on our reproduced RoboFlamingo++ codebase, maintaining all hyper-parameters for a fair comparison, except for the number of LLM layers and our proposed dynamic early-exit paradigm. Below, we detail the implementations of RoboFlamingo++ and DeeR.

## A.1  Network Architecture

For the MLLM, we utilize the pretrained model OpenFlamingo, which includes a frozen LLM and vision encoder. The vision-language fusion modules, specifically a perceiver sampler and cross-attention, are trained using the web-scraped image-text datasets LAION-2B and Multimodal C4. The architecture specifics are outlined in Table 7. For the action head, which integrates temporal information for action prediction, we employ a 4-layer LSTM to process temporal information and a 3-layer MLP for predicting actions. To mitigate overfitting, we implement dropout for the LSTM and MLP. Additionally, LayerNorm [74, 75] is applied prior to activation functions.

We configured exit points after every two self-attention layers in all MLLM models. To conserve training resources, we employed a subset of the OpenFlamingo model as our backbone. Specifically, for OpenFlamingo3B (which has 24 LLM layers) and OpenFlamingo9B (which has 32 LLM layers), we used only the first 12 layers for DeeR multiexit architecture, resulting in 6 exit points. For RoboFlamingo++, we utilize 6/12/24 LLM layers from OpenFlamingo3B and 8/16/32 LLM layers from OpenFlamingo9B to create a range of model sizes for comparing budget versus performance curve with DeeR.

Table 7: Architecture details of the OpenFlamingo models. 'xattn interval' means cross-attention interval.

| Model | Lanuguage Model | VIsion Encoder | # LLM Layers | xattn interval |
|---|---|---|---|---|
| OpenFlamingo 3B | MPT-1B (Instruct) [76] | CLIP ViT-L/14 428M | 24 | 1 |
| OpenFlamingo 9B | MPT-7B [76] | CLIP ViT-L/14 428M | 32 | 4 |

## A.2  Inference Details

After training, we obtain a multiexit model. During inference, the model remains fixed, but the computational cost of DeeR can be adjusted dynamically based on computational constraints, without modifying the multiexit model. For the 12-layer multiexit model built on top of OpenFlamingo 3B, each LLM layer consumes approximately 0.5GB of memory and requires 1.3G FLOPs during inference (with a batch size of 1, excluding the vision encoder). Given a GPU memory limit of 6GB and a maximum of 15.6G FLOPs, DeeR-B loads all 12 LLM layers, offering 6 exit points. For more resource-constrained scenarios, DeeR-S loads only the first 4 LLM layers, with 2 exit points. For the OpenFlamingo 9B model, each LLM layer consumes approximately 1.0GB of memory and requires 2.85G FLOPs. DeeR-B loads 12 LLM layers with 6 exit points, requiring about 12GB of GPU memory, while DeeR-S loads 8 LLM layers with 4 exit points, using around 8GB of memory. Crucially, users have the flexibility to define custom inference-time models beyond DeeR-S and DeeR-B by selecting the number of LLM layers to load, based on memory or FLOPs constraints.

After determining the number of LLM layers and exit points, the average FLOPs per action can be further dynamically adjusted by adjusting the exit thresholds based on the criteria in Equation (5) or Equation (6). We compute the thresholds using the validation set from environment D. However, we found that thresholds computed using just 1% of the training set achieve similar performance, demonstrating that these thresholds effectively generalize and are robust enough.

## A.3  Training Details

For both RoboFlamingo++ and the multiexit model for DeeR, we employ a two-phase training schedule. Initially, we jointly train the trainable components of the MLLM (the perceiver sampler and cross-attention layers) alongside the action head. Since the backbone MLLM is pretrained and converges more rapidly, we later freeze the MLLM and finetune only the action head, which we refer to as post-training for the action head. Our experiments indicate that this additional finetuning step for the action head results in slightly better performance. The hyperparameters used during training are detailed in Table 8. Note that the dropout rates for LSTM and MLP are 0.3 and 0.4, respectively,

except during post-training for ABCD→D, where slightly higher values of 0.4 and 0.5 are used. We report the results based on the final epoch's checkpoint, rather than selecting the best-performing checkpoint.

Table 8: Training hyper-parameters for setting D→D/ABC→D/ABCD→D.

| Hyper-parameters | Values |
|---|---|
| batch size | 4*8 |
| optimizer | AdamW |
| MLLM learning rate | 1e-4 |
| action head learning rate | 2.5e-5 |
| learninrg rate schedule | constant |
| warmup steps | 2500 |
| LSTM dropout | 0.3 |
| MLP dropout | 0.4 |
| jointly-train epochs | 4 / 4 / 3 |
| post-train epochs | 4 / 1 / 1 |
| $\lambda$ | 0.01 |
| LSTM window size | 12 |

### A.4 Training Cost

We leverage PyTorch's Automatic Mixed Precision (AMP) acceleration to optimize training efficiency. For DeeR using the OpenFlamingo 3B model, training is conducted on 8 NVIDIA V100 32G GPUs, taking approximately 14 hours for 4+4 epochs on Dataset D, 24 hours for 4+1 epochs on Dataset ABC, and 25 hours for 3+1 epochs on Dataset ABCD. When scaling DeeR to the OpenFlamingo 9B, the model is trained on 8 NVIDIA A100 80G GPUs. The training for 4+4 epochs on Dataset D takes around 24 hours. Multi-node parallel training support is included in the code, offering faster training.

## B More Experimental Results

### B.1 Quantative Results

The successful rate for the 1st to 5th tasks in a task chain is shown in Table 9, Table 10, and Table 11.

### B.2 Rollout Visualization

In Figure 6, we present a rollout visualization of DeeR successfully completing all five consecutive subtasks in a task chain.

Table 9: Detailed results in the setting D→D.

| Method | Only RGB Input | Data | No. Instructions in a Row (1000 chains) | | | | | |
|---|---|---|---|---|---|---|---|---|
| | | | 1 | 2 | 3 | 4 | 5 | Avg. len (*LLM GFLOPs*) |
| HULC | ✓ | ALL | 82.7% | 64.9% | 50.4% | 38.5% | 28.3% | 2.64 |
| SPIL | ✓ | ALL | 84.6% | 65.1% | 50.8% | 38.0% | 28.6% | 2.67 |
| RoboFlamingo | ✓ | LANG | 83.9% | 64.3% | 42.9% | 35.7% | 19.6% | 2.46 (*31.2*) |
| RoboFlamingo++ | ✓ | LANG | **87.1%** | **69.6%** | 49.6% | 37.1% | 27.2% | 2.71 (*31.2*) |
| DeeR (ours) | ✓ | LANG | 85.3% | 69.6% | 54.9% | 42.0% | 31.2% | **2.83** (*8.6*) |
| DeeR w. online (ours) | ✓ | LANG | **89.7%** | **70.5%** | 51.8% | **44.2%** | **35.3%** | **2.92** (*8.5*) |

## C Broader Impacts

The potential negative societal impacts of our method align with those typically associated with general robotic technologies. We emphasize the importance of adhering to fair and safe deployment principles in robotic systems.

Table 10: Detailed results in the setting ABCD→D.

| Method | Only RGB Input | Data | No. Instructions in a Row (1000 chains) | | | | | |
| --- | --- | --- | --- | --- | --- | --- | --- | --- |
| | | | 1 | 2 | 3 | 4 | 5 | Avg. len (*LLM GFLOPs*) |
| GR-1 | ✗ | LANG | 94.9% | 89.6% | 84.4% | 78.9% | 73.10% | 4.21 |
| HULC | ✓ | ALL | 88.9% | 73.3% | 58.7% | 47.5% | 38.3% | 3.06 |
| RT-1 | ✓ | LANG | 84.4% | 61.7% | 43.8% | 32.3% | 22.7% | 2.45 |
| RoboFlamingo | ✓ | LANG | 96.4% | 89.6% | **82.4%** | 74.0% | 66.0% | 4.08 (*31.2*) |
| RoboFlamingo++ | ✓ | LANG | **98.2%** | 87.5% | 81.8% | 73.2% | 65.5% | 4.07 (*31.2*) |
| DeeR (ours) | ✓ | LANG | **98.2%** | **90.2%** | 82.1% | **75.9%** | **67.0%** | **4.13** (*10.0*) |
| DeeR w. online (ours) | ✓ | LANG | **99.1%** | **93.3%** | 82.1% | 74.6% | 63.8% | **4.13** (*9.7*) |

Table 11: Detailed results in the setting ABC→D.

| Method | Only RGB Input | Data | No. Instructions in a Row (1000 chains) | | | | | |
| --- | --- | --- | --- | --- | --- | --- | --- | --- |
| | | | 1 | 2 | 3 | 4 | 5 | Avg. len (*LLM GFLOPs*) |
| GR-1 | ✗ | LANG | 85.4% | 71.2% | 59.6% | 49.7% | 40.1% | **3.06** |
| HULC | ✓ | ALL | 41.8% | 16.5% | 5.7% | 1.9% | 1.1% | 0.67 |
| RT-1 | ✓ | LANG | 53.3% | 22.2% | 9.4% | 3.8% | 1.3% | 0.9 |
| SPIL | ✓ | ALL | 74.2% | 46.3% | 27.6% | 14.7% | 8.0% | 1.71 |
| SuSIE | ✓ | ALL | **87.0%** | 69.0% | 49.0% | 38.0% | 26.0% | 2.69 |
| RoboFlamingo | ✓ | LANG | 82.4% | 61.9% | 46.6% | 33.1% | 23.5% | 2.47 (*31.2*) |
| RoboFlamingo++ | ✓ | LANG | 85.9% | 67.4% | 48.7% | 31.7% | 25.0% | 2.59 (*31.2*) |
| DeeR (ours) | ✓ | LANG | 86.2% | **70.1%** | **51.8%** | **41.5%** | **30.4%** | **2.82** (*12.5*) |
| DeeR w. online (ours) | ✓ | LANG | 84.8% | **72.3%** | **54.9%** | **44.6%** | **33.5%** | **2.90** (*9.5*) |


Figure 6: Visualization of DeeR rollouts of five subtasks in a task chain.

- The paper should point out any strong assumptions and how robust the results are to violations of these assumptions (e.g., independence assumptions, noiseless settings, model well-specification, asymptotic approximations only holding locally). The authors should reflect on how these assumptions might be violated in practice and what the implications would be.

- The authors should reflect on the scope of the claims made, e.g., if the approach was only tested on a few datasets or with a few runs. In general, empirical results often depend on implicit assumptions, which should be articulated.

- The authors should reflect on the factors that influence the performance of the approach. For example, a facial recognition algorithm may perform poorly when image resolution is low or images are taken in low lighting. Or a speech-to-text system might not be used reliably to provide closed captions for online lectures because it fails to handle technical jargon.

- The authors should discuss the computational efficiency of the proposed algorithms and how they scale with dataset size.

- If applicable, the authors should discuss possible limitations of their approach to address problems of privacy and fairness.

- While the authors might fear that complete honesty about limitations might be used by reviewers as grounds for rejection, a worse outcome might be that reviewers discover limitations that aren't acknowledged in the paper. The authors should use their best judgment and recognize that individual actions in favor of transparency play an impor-

tant role in developing norms that preserve the integrity of the community. Reviewers will be specifically instructed to not penalize honesty concerning limitations.

3. **Theory Assumptions and Proofs**

   Question: For each theoretical result, does the paper provide the full set of assumptions and a complete (and correct) proof?

   Answer: [NA]

   Justification: This paper does not involves theoretical result.

   Guidelines:

   - The answer NA means that the paper does not include theoretical results.
   - All the theorems, formulas, and proofs in the paper should be numbered and cross-referenced.
   - All assumptions should be clearly stated or referenced in the statement of any theorems.
   - The proofs can either appear in the main paper or the supplemental material, but if they appear in the supplemental material, the authors are encouraged to provide a short proof sketch to provide intuition.
   - Inversely, any informal proof provided in the core of the paper should be complemented by formal proofs provided in appendix or supplemental material.
   - Theorems and Lemmas that the proof relies upon should be properly referenced.

4. **Experimental Result Reproducibility**

   Question: Does the paper fully disclose all the information needed to reproduce the main experimental results of the paper to the extent that it affects the main claims and/or conclusions of the paper (regardless of whether the code and data are provided or not)?

   Answer: [Yes]

   Justification: We provide detailed information about experiments in the appendix and provide the source code that can reproduce reported results.

   Guidelines:

   - The answer NA means that the paper does not include experiments.
   - If the paper includes experiments, a No answer to this question will not be perceived well by the reviewers: Making the paper reproducible is important, regardless of whether the code and data are provided or not.
   - If the contribution is a dataset and/or model, the authors should describe the steps taken to make their results reproducible or verifiable.
   - Depending on the contribution, reproducibility can be accomplished in various ways. For example, if the contribution is a novel architecture, describing the architecture fully might suffice, or if the contribution is a specific model and empirical evaluation, it may be necessary to either make it possible for others to replicate the model with the same dataset, or provide access to the model. In general. releasing code and data is often one good way to accomplish this, but reproducibility can also be provided via detailed instructions for how to replicate the results, access to a hosted model (e.g., in the case of a large language model), releasing of a model checkpoint, or other means that are appropriate to the research performed.
   - While NeurIPS does not require releasing code, the conference does require all submissions to provide some reasonable avenue for reproducibility, which may depend on the nature of the contribution. For example
     (a) If the contribution is primarily a new algorithm, the paper should make it clear how to reproduce that algorithm.
     (b) If the contribution is primarily a new model architecture, the paper should describe the architecture clearly and fully.
     (c) If the contribution is a new model (e.g., a large language model), then there should either be a way to access this model for reproducing the results or a way to reproduce the model (e.g., with an open-source dataset or instructions for how to construct the dataset).

(d) We recognize that reproducibility may be tricky in some cases, in which case authors are welcome to describe the particular way they provide for reproducibility. In the case of closed-source models, it may be that access to the model is limited in some way (e.g., to registered users), but it should be possible for other researchers to have some path to reproducing or verifying the results.

5. **Open access to data and code**

Question: Does the paper provide open access to the data and code, with sufficient instructions to faithfully reproduce the main experimental results, as described in supplemental material?

Answer: [Yes]

Justification: We will release of code and data.

Guidelines:

- The answer NA means that paper does not include experiments requiring code.
- Please see the NeurIPS code and data submission guidelines (`https://nips.cc/public/guides/CodeSubmissionPolicy`) for more details.
- While we encourage the release of code and data, we understand that this might not be possible, so "No" is an acceptable answer. Papers cannot be rejected simply for not including code, unless this is central to the contribution (e.g., for a new open-source benchmark).
- The instructions should contain the exact command and environment needed to run to reproduce the results. See the NeurIPS code and data submission guidelines (`https://nips.cc/public/guides/CodeSubmissionPolicy`) for more details.
- The authors should provide instructions on data access and preparation, including how to access the raw data, preprocessed data, intermediate data, and generated data, etc.
- The authors should provide scripts to reproduce all experimental results for the new proposed method and baselines. If only a subset of experiments are reproducible, they should state which ones are omitted from the script and why.
- At submission time, to preserve anonymity, the authors should release anonymized versions (if applicable).
- Providing as much information as possible in supplemental material (appended to the paper) is recommended, but including URLs to data and code is permitted.

6. **Experimental Setting/Details**

Question: Does the paper specify all the training and test details (e.g., data splits, hyper-parameters, how they were chosen, type of optimizer, etc.) necessary to understand the results?

Answer: [Yes]

Justification: We specify all the training and test details in the main text, appendix, and source code.

Guidelines:

- The answer NA means that the paper does not include experiments.
- The experimental setting should be presented in the core of the paper to a level of detail that is necessary to appreciate the results and make sense of them.
- The full details can be provided either with the code, in appendix, or as supplemental material.

7. **Experiment Statistical Significance**

Question: Does the paper report error bars suitably and correctly defined or other appropriate information about the statistical significance of the experiments?

Answer: [Yes]

Justification: We repeated experiments with several runnings and found error bar is relatively small.

Guidelines:

- The answer NA means that the paper does not include experiments.
- The authors should answer "Yes" if the results are accompanied by error bars, confidence intervals, or statistical significance tests, at least for the experiments that support the main claims of the paper.
- The factors of variability that the error bars are capturing should be clearly stated (for example, train/test split, initialization, random drawing of some parameter, or overall run with given experimental conditions).
- The method for calculating the error bars should be explained (closed form formula, call to a library function, bootstrap, etc.)
- The assumptions made should be given (e.g., Normally distributed errors).
- It should be clear whether the error bar is the standard deviation or the standard error of the mean.
- It is OK to report 1-sigma error bars, but one should state it. The authors should preferably report a 2-sigma error bar than state that they have a 96% CI, if the hypothesis of Normality of errors is not verified.
- For asymmetric distributions, the authors should be careful not to show in tables or figures symmetric error bars that would yield results that are out of range (e.g. negative error rates).
- If error bars are reported in tables or plots, The authors should explain in the text how they were calculated and reference the corresponding figures or tables in the text.

8. **Experiments Compute Resources**

Question: For each experiment, does the paper provide sufficient information on the computer resources (type of compute workers, memory, time of execution) needed to reproduce the experiments?

Answer: [Yes]

Justification: We provided sufficient information on the computer resources in the main text and appendix.

Guidelines:

- The answer NA means that the paper does not include experiments.
- The paper should indicate the type of compute workers CPU or GPU, internal cluster, or cloud provider, including relevant memory and storage.
- The paper should provide the amount of compute required for each of the individual experimental runs as well as estimate the total compute.
- The paper should disclose whether the full research project required more compute than the experiments reported in the paper (e.g., preliminary or failed experiments that didn't make it into the paper).

9. **Code Of Ethics**

Question: Does the research conducted in the paper conform, in every respect, with the NeurIPS Code of Ethics https://neurips.cc/public/EthicsGuidelines?

Answer: [Yes]

Justification: the research conducted in the paper conformed, in every respect, with the NeurIPS Code of Ethics.

Guidelines:

- The answer NA means that the authors have not reviewed the NeurIPS Code of Ethics.
- If the authors answer No, they should explain the special circumstances that require a deviation from the Code of Ethics.
- The authors should make sure to preserve anonymity (e.g., if there is a special consideration due to laws or regulations in their jurisdiction).

10. **Broader Impacts**

Question: Does the paper discuss both potential positive societal impacts and negative societal impacts of the work performed?

Answer: [Yes]

Justification: discuss both potential positive societal impacts and negative societal impacts of the work in the appendix.

Guidelines:

- The answer NA means that there is no societal impact of the work performed.
- If the authors answer NA or No, they should explain why their work has no societal impact or why the paper does not address societal impact.
- Examples of negative societal impacts include potential malicious or unintended uses (e.g., disinformation, generating fake profiles, surveillance), fairness considerations (e.g., deployment of technologies that could make decisions that unfairly impact specific groups), privacy considerations, and security considerations.
- The conference expects that many papers will be foundational research and not tied to particular applications, let alone deployments. However, if there is a direct path to any negative applications, the authors should point it out. For example, it is legitimate to point out that an improvement in the quality of generative models could be used to generate deepfakes for disinformation. On the other hand, it is not needed to point out that a generic algorithm for optimizing neural networks could enable people to train models that generate Deepfakes faster.
- The authors should consider possible harms that could arise when the technology is being used as intended and functioning correctly, harms that could arise when the technology is being used as intended but gives incorrect results, and harms following from (intentional or unintentional) misuse of the technology.
- If there are negative societal impacts, the authors could also discuss possible mitigation strategies (e.g., gated release of models, providing defenses in addition to attacks, mechanisms for monitoring misuse, mechanisms to monitor how a system learns from feedback over time, improving the efficiency and accessibility of ML).

11. **Safeguards**

Question: Does the paper describe safeguards that have been put in place for responsible release of data or models that have a high risk for misuse (e.g., pretrained language models, image generators, or scraped datasets)?

Answer: [Yes]

Justification: We describe safeguards for responsible release of models in the social impacts section.

Guidelines:

- The answer NA means that the paper poses no such risks.
- Released models that have a high risk for misuse or dual-use should be released with necessary safeguards to allow for controlled use of the model, for example by requiring that users adhere to usage guidelines or restrictions to access the model or implementing safety filters.
- Datasets that have been scraped from the Internet could pose safety risks. The authors should describe how they avoided releasing unsafe images.
- We recognize that providing effective safeguards is challenging, and many papers do not require this, but we encourage authors to take this into account and make a best faith effort.

12. **Licenses for existing assets**

Question: Are the creators or original owners of assets (e.g., code, data, models), used in the paper, properly credited and are the license and terms of use explicitly mentioned and properly respected?

Answer: [Yes]

Justification: We properly credited the creators or original owners of assets (e.g., code, data, models), used in the paper and conformed the license and terms.

Guidelines:

- The answer NA means that the paper does not use existing assets.
- The authors should cite the original paper that produced the code package or dataset.

- The authors should state which version of the asset is used and, if possible, include a URL.
- The name of the license (e.g., CC-BY 4.0) should be included for each asset.
- For scraped data from a particular source (e.g., website), the copyright and terms of service of that source should be provided.
- If assets are released, the license, copyright information, and terms of use in the package should be provided. For popular datasets, `paperswithcode.com/datasets` has curated licenses for some datasets. Their licensing guide can help determine the license of a dataset.
- For existing datasets that are re-packaged, both the original license and the license of the derived asset (if it has changed) should be provided.
- If this information is not available online, the authors are encouraged to reach out to the asset's creators.

13. **New Assets**

Question: Are new assets introduced in the paper well documented and is the documentation provided alongside the assets?

Answer: [Yes]

Justification: We communicated the details of the dataset/code/model as part of their submission.

Guidelines:

- The answer NA means that the paper does not release new assets.
- Researchers should communicate the details of the dataset/code/model as part of their submissions via structured templates. This includes details about training, license, limitations, etc.
- The paper should discuss whether and how consent was obtained from people whose asset is used.
- At submission time, remember to anonymize your assets (if applicable). You can either create an anonymized URL or include an anonymized zip file.

14. **Crowdsourcing and Research with Human Subjects**

Question: For crowdsourcing experiments and research with human subjects, does the paper include the full text of instructions given to participants and screenshots, if applicable, as well as details about compensation (if any)?

Answer: [NA]

Justification: Our paper does not involve study participants.

Guidelines:

- The answer NA means that the paper does not involve crowdsourcing nor research with human subjects.
- Including this information in the supplemental material is fine, but if the main contribution of the paper involves human subjects, then as much detail as possible should be included in the main paper.
- According to the NeurIPS Code of Ethics, workers involved in data collection, curation, or other labor should be paid at least the minimum wage in the country of the data collector.

15. **Institutional Review Board (IRB) Approvals or Equivalent for Research with Human Subjects**

Question: Does the paper describe potential risks incurred by study participants, whether such risks were disclosed to the subjects, and whether Institutional Review Board (IRB) approvals (or an equivalent approval/review based on the requirements of your country or institution) were obtained?

Answer: [NA]

Justification: Our paper does not involve study participants.

Guidelines:

- The answer NA means that the paper does not involve crowdsourcing nor research with human subjects.
- Depending on the country in which research is conducted, IRB approval (or equivalent) may be required for any human subjects research. If you obtained IRB approval, you should clearly state this in the paper.
- We recognize that the procedures for this may vary significantly between institutions and locations, and we expect authors to adhere to the NeurIPS Code of Ethics and the guidelines for their institution.
- For initial submissions, do not include any information that would break anonymity (if applicable), such as the institution conducting the review.

