# OpenReview forum: "DeeR-VLA: Dynamic Inference of Multimodal Large Language Models for Efficient Robot Execution"
_NeurIPS.cc/2024/Conference — NeurIPS 2024 poster_

### Official Review · Reviewer_Cni2 · 2024-07-08

**Soundness:** 3
**Presentation:** 3
**Contribution:** 3
**Rating:** 7
**Confidence:** 4

**Summary:**

This paper aims to reduce computation time and enhance inference efficiency when deploying Multimodal Large Language Models on real robots. It introduces a dynamic neural network that selectively activates only a small portion of the network while maintaining performance levels comparable to the original network. The author proposes a straightforward yet effective early exiting criterion that compares hidden states from various attention blocks. To tackle training challenges, the author incorporates all hidden states into the loss functions. Experimental results demonstrate improvements in computational and memory efficiency while achieving similar performance to the original model.

**Strengths:**

1. The Dynamic Early-Exit for Robotic MLLM (DeeR) framework utilizes dynamic networks with the early exiting strategy to significantly reduce computational and memory requirements, making it feasible to run complex models on resource-limited robotic platforms.
2. This paper conducts comprehensive evaluations in the simulation environments. It shows that DeeR can not only get competitive performance with less computation but also has great scalability by dynamically adjusting the size of the activated model.

**Weaknesses:**

1. The author shows that developing MLLMs for real robots is challenging and claims that this framework can get efficient robot execution, but the experiments are conducted in simulated environments.
2. This framework relies on threshold determination. The process of setting optimal thresholds for early exits requires careful tuning and may involve complex optimization problems. Although the authors show its capability in low-data environments and unseen situations, combined with significant changes to the architecture and training process, it might still be challenging for some applications in the real world.
3. The training is conducted using a limited subset of language-annotated data, which might not capture the full diversity of real-world scenarios.

**Questions:**

1. Looks like using dynamic networks with the early exiting strategy is a common strategy in efficient LLMs? It's better to show more about it in related work and highlight your contributions and novelty.
2. While auxiliary losses improve performance marginally, they should have additional complexity to the training process. It's better to show more results and analysis of these ablations.

**Limitations:**

1. As shown in the weakness section, some components of this framework will bring challenges to the implementations in the real robot. It's better to show more real-world experiments or more analysis which can show that it can be effective in the real world.
2. The experiments in simulation environments are comprehensive but some analyses are not very clear. For example, how to know this framework can solve thresholds effectively in challenging scenarios such as low-data environments and generalization to unseen situations according to Table 2 with successful length? Also, it's better to get more analysis for the auxiliary losses.

---

> ### Author Rebuttal · Authors · 2024-08-07
>
> Thank you for your time and constructive feedback. Please see the response below.
>
> **W1/L1**: `Experiments are limited in simulated environments `; `show more analysis which can show that it can be effective in the real world `
>
> Thanks. We admit that the submitted version of manuscript limits in the simulated CALVIN benchmark. We are actively expanding DeeR's application to real-world robotic tasks by integrating it with the OpenVLA framework, which is tailored for real-world manipulation, rather than RoboFlamingo.  However, training on the Open X-Embodiment Dataset [1] is time-intensive, and setting up a real evaluation platform using the Franka robot arm is a significant workload. Due to the tight timeline of this rebuttal, we regret that comprehensive results are not yet available.
>
> Nevertheless, we have initial results on realistic inference times evaluated on an NVIDIA V100 GPU. **DeeR achieves a 68.1% reduction in LLM inference time compared to RoboFlamingo++, which aligns with the theoretical 80.7% reduction in FLOPS.** This was accomplished without specific optimization for early-exit implementation, and we expect future optimization to further improve DeeR’s operational efficiency.
>
> |ABCD->D|Avg Len|Avg LLM GLOPS|Avg LLM inference time|
> |-|-|-|-|
> |RoboFlamingo++|4.07|31.2|55ms|
> |DeeR|4.08|6.0|17.5ms|
>
> Further, we wish to highlight that the **CALVIN benchmark are developed for capturing many challenges present in real-world settings.** It encompasses a wide array of robotic manipulation tasks and features over 400 crowd-sourced natural language instructions, posing significant challenges to SOTA algorithms due to its realistic complexity.
>
> We believe these points collectively demonstrate DeeR's substantial potential for real-world applications.
>
> **W2/L2**: `Setting optimal thresholds for early exits requires careful tuning. Can this framework solve thresholds effectively in challenging low-data or unseen situations? `
>
> Thank you for raising this issue. We would like to point out that DeeR determine optimal thresholds using a specialized algorithm rather than relying on manual trial-and-error methods, As detailed in lines 239-260. To validate the effectiveness of threshold-solving framework in real-world-like scenarios, including low-data environments and situations unseen during training, we conducted further experiments. Given a predined GLOPS budget, we solve thresholds directly using a validation set D or **a 0.1% subset of the training data ABC (low-data and unseen situation)**, then assess the model under these thresholds. The results are as follows:
>
> |#data|avg. len (ABC->D)|GLOPS|
> |-|-|-|
> |validation|2.82|12.5|
>  |0.1% training|2.80|12.5|
>
> These experiments demonstrate that our computed thresholds can effectively generalize and be capable of handling challenges typical of real-world applications.
>
> **W3**:  `The training is conducted using a limited subset of language-annotated data, which might not capture the full diversity of real-world scenarios. `
>
> Thanks for your insightful comment. I would like to clarify that **our early-exit strategy for efficient robot execution is not dependent on language-annotated data**. The essence of our method lies in dynamically adapting the complexity of the robotic policy based on the specific circumstances. **This policy could be a MLLM or a purely visual model**, independent of language annotations. In this manuscript, we opted for an MLLM policy due to its demonstrated effectiveness in robotic learning such as RT-2.
>
> **Q1**: `using dynamic networks with the early exiting strategy is a common strategy in efficient LLMs? `; ` show more about it in related work and highlight your contributions and novelty. `
>
> Thank you for your comment. To the best of our knowledge, while early exiting is a recognized strategy in computer vision, its application in LLMs is not fully explored, with only a few related works such as LayerSkip [1] and EELLM [2].
>
> In response to your suggestion, we delineate our unique contributions as follows:
> 1. We first **identified MLLM redandency in robotic scenarios** and find that 'easier' circumstances, the majority of scenario, can be efficiently handled by smaller models than typical MLLM (Table 1 of our manuscript).
> 2. To the best of our knowledge, **DeeR is first  to introduce a novel early-exit strategy specially designed for robotic MLLM policy, addressing significant challenges in applying early-exit to action generation**:
>   - **Metric for Early Exiting**: DeeR employs an action consistency metric instead of typical softmax confidence or entropy, crucial for action generation where direct softmax outputs are impractical.
>   - **Threshold Optimization**: Previous approaches often relied on manually tuning thresholds or solving threshold on datasets. DeeR uses online learning algorithms to dynamically set thresholds during interactions, enhancing early-exit performance (see Table 2, last row).
>   - **Mitigating Training/dynamic Inference Discrepancy in Temporal Integration**: DeeR addresses this by introducing a loss function that randomly samples exits at each timestep, enhancing robustness across dynamic inferences (lines 268-281).
>
> We will expand on these points in the related works of our revision.
>
> **Q2**: `Additional complexity of auxiliary loss `
>
> We admit that the impact of the auxiliary loss on performance is modest. However, **we retain the auxiliary loss due to its little additional training cost and no extra inference cost**, improving the performance for almost free.
>
> |Tested on V100|w.o. AuxLoss|w. AuxLoss|
>  |-|-|-|
> |train step/second|1.18|1.07|
>
>  [1] Elhoushi, Mostafa, et al. "Layer skip: Enabling early exit inference and self-speculative decoding." arXiv preprint arXiv:2404.16710.
>
> [2] Chen, Yanxi, et al. "Ee-llm: Large-scale training and inference of early-exit large language models with 3d parallelism." arXiv:2312.04916.

---

> > ### Comment · Reviewer_Cni2 · 2024-08-12
> > **Thanks for your response!**
> >
> > Thanks for the rebuttal! I appreciate the justification for the optimal threshold and your contributions. Your other explanations also mostly address my concern. Although it still hasn't been proven effective in the real robot setting, it has great performance in challenging benchmarks and has potential to solve this challenging problem. I'd like to increase my rating

---

> > > ### Author Response · Authors · 2024-08-13
> > >
> > > Thanks for taking the time to review our paper and for your prompt response. We also appreciate your constructive suggestions, which help us more clearly identify  the effectiveness of the solved threshold and the contributions of this paper. We are honored to have received your support for its acceptance.

---

### Official Review · Reviewer_4zeN · 2024-07-11

**Soundness:** 2
**Presentation:** 3
**Contribution:** 2
**Rating:** 5
**Confidence:** 4

**Summary:**

The paper presents a framework called Dynamic Early-Exit for Robotic MLLM (DeeR) aimed at improving the computational efficiency of Multimodal Large Language Models (MLLMs) used in robotic applications. The DeeR framework tries to address the challenge of deploying MLLMs on robots with limited computational resources by dynamically adjusting the size of the activated MLLM based on the complexity of the task. The approach leverages a multi-exit architecture, allowing the model to terminate processing early when a smaller model is sufficient for the task at hand.

**Strengths:**

1. The paper introduces an approach to dynamically adjust the size of MLLMs based on the complexity of the task, leveraging a multi-exit architecture. This approach addresses a critical challenge in deploying MLLMs on resource-constrained robotic platforms.

2. The proposed approach seems relevant to the field of robotics, where real-time performance and efficient resource utilization are critical. The ability to dynamically adjust the model size based on task complexity is a practical solution for real-world robotic applications.

3. DeeR seems effectively reduces computational costs and GPU memory usage, making it feasible to deploy MLLMs on real-world robots with limited computational resources.

**Weaknesses:**

1. While the paper demonstrates the effectiveness of DeeR on the CALVIN robot manipulation benchmark, it is unclear how well the approach generalizes to other robotic tasks and environments. Additional experiments on a wider range of tasks would strengthen the paper.

2. The paper compares DeeR to other state-of-the-art methods, but it would benefit from a more detailed comparison with other efficient LLM approaches, including those focused on model compression and efficient structural design.

**Questions:**

1. How well does DeeR generalize to other robotic tasks beyond the CALVIN robot manipulation benchmark? Are there any limitations when applying DeeR to different tasks?

2. Can the authors provide more details on the implementation of the multi-exit architecture and the early-termination criteria?

3. How does DeeR compare to other efficient LLM approaches, such as model compression and efficient structural design, in terms of computational efficiency and performance?

**Limitations:**

1. The effectiveness of DeeR may depend on task-specific optimizations, and the approach may require tuning for different robotic tasks and environments.

2. The scalability of DeeR to larger and more complex robotic systems is not fully explored in the paper. Further research is needed to assess the scalability of the approach.

---

> ### Author Rebuttal · Authors · 2024-08-07
>
> Thank you for your time and constructive feedback. Please see the response below.
>
> **W1/Q1**: `How well the approach generalizes to other robotic tasks and environments?  `; `Are there any limitations when applying DeeR to different tasks?`
>
> In principle, DeeR is designed for general robotic tasks with action outputs, and it can be straightforwardly applied to other tasks and environments. Implementation only involves integrating our early-exit strategy into an existing robotic policy network for the task through multi-exit adaptation. Applying DeeR to additional tasks like navigation and dexterous manipulation is to be explored for future research.
>
> One potential limitation when adapting DeeR to diverse tasks involves calculating action consistency metrics for termination. In tasks that combine different types of actions, such as thw whole body movement and arm manipulation, where certain actions may dominate, a simple L2 distance may not accurately reflect consistency. Addressing this issue is a focus for our future work.
>
> **W2/Q3**: `Comparison with other efficient LLM approaches such as model compression and efficient structural design `
>
> Thank you for your suggestion. It's important to note that model compression techniques like quantization and pruning, as well as efficient structural designs, are orthogonal to early-exit strategies like DeeR. These approaches enhance efficiency from distinct aspects. For instance, quantization reduces memory requirements by decreasing the precision of each parameter, while early-exit strategies like DeeR by dynamically skipping unnecessary computation in deeper layers. We take quantization as the representative example to show DeeR can integrate with such techniques:
>
> |DeeR|Mem|Avg len (ABCD->D)|
> |-|-|-|
> |float32|6G|4.13|
> |bf16|3G|4.11|
> |int4|1.7G|3.91|
>
> In future, we plan to further enhance the efficiency of our LLM-based robotic systems by integrating DeeR with both pruning and structural optimizations.
>
> **Q2**: `more details on the implementation of the multi-exit architecture and the early-termination criteria `
>
> Thank you for your inquiry. Here are more detailed implementations:
> - **Multi-exit Architecture**: In our approach, the layers of the LLM are segmented into multiple exits at uniform intervals—every two layers in our implementation. At any intermediate exit $i$, we employ a max-pooling operator to aggregate information across the token dimension, serving as the output feature for that exit. During inference, all exits share a single action head for action prediction. The action head includes a lightweight 4-layer LSTM and a 3-layer MLP.
> - **Early-Termination Criteria**: Our termination criteria rely on the consistency of action predictions between consecutive exits. Specifically, if the L2 distance between the action predictions at exits $i$ and $i-1$ is small, we terminate at exit $i$. For the first exit, the input feature of that layer is used as the previous feature for comparison.
>
> Feel free to ask for further details on these aspects.
>
> **L1**: `The effectiveness of DeeR may depend on task-specific optimizations, and the approach may require tuning for different robotic tasks and environments. `
>
> Thank you for your comment. DeeR is designed for general manipulation tasks with continuous action outputs and was applied across three CALVIN settings without task-specific optimizations. Adapting to significantly different environments or tasks, like with many existing deep learning methods, often presents challenges that may be attributed to data limitations. In foundational models, scaling up the data and its diversity can often mitigate the need for extensive parameter tuning and task-specific optimization. Addressing data limitations within embodied intelligence remains a compelling direction for future research in the community.
>
> **L2**: `The scalability of DeeR to larger and more complex robotic systems is not fully explored in the paper. Further research is needed to assess the scalability of the approach. `
>
> Thank you for highlighting the direction of future research. While the CALVIN benchmark is complex and challenging within simulated environments, we acknowledge the necessity to explore the scalability of DeeR in more diverse and realistic robotic systems.  Actually, we are actively expanding DeeR to larger and more complex real-world robotic tasks by integrating it with the OpenVLA framework, which is tailored for real-world manipulation and training on the Open X-Embodiment Dataset. Due to the tight timeline of this rebuttal, we regret that comprehensive results are not yet available. But we believe since OpenVLA is already effective as an MLLM policy for real robot manipulation, incorporating our early-exit strategy should be straightforward.
>
> **Conclusion**
>
> In conclusion, we appreciate the opportunity to address these vital concerns. We hope that our responses sufficiently clarify your question. If there are any additional questions about our research, please do not hesitate to reach out.

---

> > ### Author Response · Authors · 2024-08-12
> >
> > Dear Reviewer 4zeN,
> >
> > Thank you for your constructive review and positive comment on our paper. We appreciate the opportunity to address your concerns.
> >
> > We hope that our responses sufficiently clarify your questions, regarding the comparison with other efficient LLM approaches and the scalability of DeeR. With the deadline just two days away, we humbly request your prompt feedback to enable further discussion and revisions if needed.
> >
> > Best regards,
> >
> > The Authors

---

### Official Review · Reviewer_LFFp · 2024-07-12

**Soundness:** 3
**Presentation:** 3
**Contribution:** 3
**Rating:** 7
**Confidence:** 4

**Summary:**

DeeR provides an effective solution to the computational inefficiencies of MLLMs in robotic applications, enabling their use in resource-constrained environments while maintaining high performance. The paper emphasizes the practical implications of this approach, making advanced robotic capabilities more accessible and efficient.

**Strengths:**

1.The DeeR model achieves competitive performance compared to state-of-the-art (SOTA) models like GR-1 and RoboFlamingo, especially in generalization scenarios (ABC→D). And it significantly reduces computational requirements, needing 5.9x fewer average FLOPs and 2x less GPU memory while maintaining high performance.
2.The model demonstrates good scalability, reducing computation by 1.8-5.7x and peak FLOPs and memory usage by 2.7x-4.0x without sacrificing performance when scaled up using OpenFlamingo 9B.
3.The document provides a comprehensive evaluation, including ablation studies on auxiliary losses and exit criteria, highlighting the importance of these components for the model's performance.

**Weaknesses:**

1.In section 3.3, authors mentioned the discrepancy between training and inference. I’m not sure about how the training method solved this discrepancy. Because every layer’s exit are uniformly sampled during the training process, which is totally different from the way in inference. And I also concern this way could affect the capability of LLM, because the output of every layer would lean to similar distribution. So why not directly use a small policy network, like rt-1.
2.The function of auxiliary loss seems to be less significant. According to Table 3, the improvements are limited (0.06~0.08). And why still need auxiliary action head under the sampling way mentioned above , does it really matter? Besides, does the parameters of action head module and aux. head module are shared?
3.Why DeeR can outperform RoboFlamingo with lower computation? More detailed analysis about this would be better for presenting the highlight of DeeR.
4.The main idea of this paper seems to be similar with Layer skip proposed by Meta. Can you explain the main difference between them?
5.Although this work can reduce computation cost and usage of GPU memory so much, I also care about the true inference time/frequency in a real environment?

**Questions:**

Please refer to the weakness.

---

> ### Author Rebuttal · Authors · 2024-08-07
>
> Thank you for your time and feedback. Please see the response below.
>
> **W1**: `uniformly sampling  is totally different from the way in inference`
>
> We recognize that uniform sampling during training does not perfectly replicate dynamic inference conditions. However, we believe this approach is relatively suitable for dynamic exit for several reasons:
> - DeeR functions as a dynamic early exiting system where **exit probabilities adjust based on the difficulty of the current scenario and computational constraints. These probabilities are inherently unpredictable.** Under limited resources, DeeR may early exit in the way like '11121'; conversely, it might be '66666'.
> - Uniform sampling exposes the model to a broad spectrum of scenarios, preparing it for any potential exit sequence during dynamic inference and **ensuring it can handle all possible combinations and sequences of exits, including extreme cases**.
>
> Nevertheless,  exploring alternative sampling distributions can be future work.
>
> **W1**: `Concerns About Similar Layer Output Distributions and Impact on LLM Capability`; `Why not directly use a small policy network, like rt-1`
>
> Thank you for your comments.  If trained from scratch with uniform sampling across layers, this degradation may occur. However, our approach effectively mitigates this risk. **By adopting the Flamingo methodology, we keep LLM parameters fixed and only fine-tune the cross-attention mechanisms, ensuring stable core language knowledge across layers.**
>
>  OpenFlamingo progressively enhances representation from one layer to the next. Our DeeR model, with its fixed LLM layers and general knowledge, retains this progressive enhancement, ensuring varied contributions from each layer and preventing distribution convergence. We confirmed this by comparing action inference using feature  at the 2nd and 12th layers:
>
> |D->D|Avg Len|
> |-|-|
> |L=2|2.03|
> |L=12|2.84|
>
> This notable improvement in deeper layers also underscores the benefits of using a larger MLLM model over a simpler alternative like RT-1. By utilizing a pretrained MLLM, DeeR leverages extensive knowledge, learning critical representations for robotic tasks, as evidenced in works like RT-2.
>
> **W2**:  `Why still need auxiliary action head, does it really matter?`;  `Does the parameters of action head and aux. head are shared? `
>
> We acknowledge that the impact of the auxiliary loss on performance is modest. However, **we retain the auxiliary loss due to its little additional training cost and no extra inference cost**, improving performance for almost free.
>
> |Tested on V100|w.o. AuxLoss|w. AuxLoss|
>  |-|-|-|
> |train step/second|1.18|1.07|
>
> Action head and auxiliary head are not shared. The auxiliary head is discarded after training.
>
> **W3**: `Why DeeR can outperform RoboFlamingo ?`
>
> Thanks for your question. DeeR builds upon RoboFlamingo++ and maintains consistent architecture and hyperparameters. The key distinction during training is DeeR's use of the loss function $\mathcal{L}^*$ and an auxiliary loss $\mathcal{L}_\text{aux}$ (detailed in section 3.3). To better understand the performance gains, we conducted an ablation study.
>
> It's important to note that variants without $\mathcal{L}^*$ do not support dynamic exit. Therefore, we evaluated all results at the fixed last exit to assess the upper bound performance of the trained model. While these two losses are initially designed to enable effective early inference, we observe that they also marginally enhance overall performance. We conjecture that this improvement may be due to the introduction of explicit supervision at shallow network layers, making the training process more effective,  as seen in similar studies [1]. We will make the ablation clearer in the revision.
>
> ||$$\mathcal{L}^*$$|$$\mathcal{L}_\text{aux}$$|Best Len (D->D)|
> |-|-|-|-|
> |DeeR|✔|✔|2.84|
> ||✔||2.80|
> |||✔|2.74|
> |Flamingo++|||2.71|
>
> **W4**: `Comparison with LayerSkip`
>
> Our work and LayerSkip both utilize early exit strategies, but DeeR significantly differs in several ways, as outlined in lines 116-121 of our paper:
> 1. **Dynamic vs. Static Early Exit**: LayerSkip uses a static approach, exiting at pre-specified layers for all tokens. Conversely, **DeeR dynamically determines exit points based on the complexity of the current scenario**, employing a novel termination metric based on action consistency and algorithmically set thresholds according to computational budgets.
> 2. **Action Generation vs. Language Generation**: While LayerSkip is tailored for language generation, DeeR is designed for action generation, which necessitates integrating temporal information for decision. This integration introduces challenges like training-inference discrepancies, which DeeR addresses by employing a loss function that samples across different exits at each timestep.
>
> **W5**: `True inference time/frequency in a real environment`
>
> Thanks. We admit that the submitted version of manuscript limits in the simulated CALVIN benchmark. We are actively expanding DeeR's application to real-world robotic tasks by integrating it with the OpenVLA framework, which is tailored for real-world manipulation, rather than RoboFlamingo.  Due to the tight timeline of this rebuttal, comprehensive results are not yet available. But we believe since OpenVLA is already effective as an MLLM policy for real robot manipulation, incorporating our early-exit strategy should be straightforward.
>
> Currently, we have initial results on realistic inference times evaluated on an NVIDIA V100 GPU. **DeeR achieves a 68.1% reduction in LLM inference time compared to RoboFlamingo++, which aligns with the theoretical 80.7% reduction in FLOPS.**
>
> |ABCD->D|Avg Len|Avg LLM GLOPS|Avg LLM inference time|
> |-|-|-|-|
> |RoboFlamingo++|4.07|31.2|55ms|
> |DeeR|4.08|6.0|17.5ms|
>
> We believe these points collectively demonstrate DeeR's substantial potential for real-world applications.
>
> [1] Zhao, Hengshuang, et al. "Pyramid scene parsing network." CVPR. 2017.

---

> > ### Author Response · Authors · 2024-08-12
> >
> > Dear Reviewer LFFp,
> >
> > Thank you for your constructive review and positive feedback on our paper.
> >
> > We believe that the rebuttal may address your concerns such as the uniform sampling strategy, true inference time, and comparison with Layerskip. Given the deadline around 2 days, we would appreciate your prompt feedback to ensure any further discussion and revisions can be made timely.
> >
> > Thank you for your time and consideration again.
> >
> > Best regards,
> >
> > authors

---

> > > ### Comment · Reviewer_LFFp · 2024-08-13
> > > **Official Comment by Reviewer LFFp**
> > >
> > > Thanks authors for the response. As to W4, I think the DeeR shares a similar motivation with Layerskip, which is a dynamic inference according to the input scenarios (Please refer to Figure2 in Layerskip). It weakens the overall contribution of this paper.  Considering the other concern has also been solved and this work solves a vital problem in manipulations, I'd like to keep my score to 5.
> > >
> > > [1] LayerSkip: Enabling Early Exit Inference and Self-Speculative Decoding.

---

> ### Author Response · Authors · 2024-08-13
>
> We really appreciate your prompt response. We wish to clarify our contributions in comparison to Layerskip.
>
> Although Layerskip's Figure 2 intuitively demonstrates that the optimal minimum layer for predicting the current token may vary, developing methods to dynamically determine the best layer to exit during inference remains a challenge. **As illustrated in Sections 4.2 and Figures 6, 8, and 10 of Layerskip, it opts to statically select a fixed layer for exiting. Furthermore, the conclusion of Layerskip suggests exploring "dynamic conditions to determine a different exit layer for each token to improve the token acceptance rate of self-speculative decoding" in the future**. This highlights that our approach of dynamically exiting is a unique contribution of DeeR, which develops the action consistency metrics, the threshold solving algorithm, and training strategy for enabling dynamic exit.
>
> We hope this clarification help you understand DeeR. Please do not hesitate to contact us if you have any further questions. Thank you once again for your time!

---

> > ### Comment · Reviewer_LFFp · 2024-08-14
> > **Official Comment by Reviewer LFFp**
> >
> > Thanks for your explanation.
> >
> > I think the future design in the conclusion of Skiplayer emphasizes the output token.
> >
> > Interestingly, the backbone of Roboflamingo only infers/decodes once and then synthesizes a token through average pooling, which is then input into the action MLP. Therefore, it can be understood as operating on just one output token.
> >
> > From my perspective, if Skiplayer were to be applied to roboflamingo, there would be no need for additional future dynamic operations, since the backbone of roboflamingo only requires one inference/decoding.
> >
> > Looking forward to your discussion！

---

> ### Author Response · Authors · 2024-08-14
>
> Thanks for your patience!
>
>  You've rightly differentiated between text generation, which often produces multiple tokens, and action generation, where the model extracts a single token representation per language instruction and current image $x_t$. **However, it's important to note that decision-making/control is temporal, requiring dynamic inference across timesteps and tasks.**
>
> For example, consider two robot tasks—one simpler, one more complex—each requiring about 100 action steps. When Skiplayer is applied to Roboflamingo, it generates one token per action step across both tasks, **using fixed LLM layers for all 200 time steps**, for instance, L=9. Setting a much smaller L, such as L=2, might not provide sufficient capacity for the more complex task. In contrast, with DeeR, the layers is used dynamically according to the task and specific timestep within the task—e.g., 12235775... for the simpler task and 5678996... for the more complex one. This dynamic inference way along timesteps and tasks allows DeeR the potential to further reduce computational costs.
>
> We appreciate this opportunity to address your concerns. We hope our responses have clarified your questions. If you have any further inquiries about our research, please feel free to reach out.

---

### Official Review · Reviewer_4SAr · 2024-07-13

**Soundness:** 3
**Presentation:** 3
**Contribution:** 3
**Rating:** 7
**Confidence:** 5

**Summary:**

This article explores the application of large language models in robotic manipulation tasks from a very interesting perspective. The authors present a notable observation: the majority of procedures involved in robot control for diverse task execution comprise relatively simple scenarios, which can be effectively managed using substantially smaller models to generate appropriate robotic actions. Therefore, this paper designed an early-exit mechanism to improve the inference efficiency of large language models.

**Strengths:**

This article presents a very interesting research motivation, identifying computational redundancy in large language models applied to robotic manipulation, and proposing an early exit mechanism to improve operational efficiency. The authors discussed and carefully designed the early exit mechanism.

**Weaknesses:**

1. The evaluation is only conducted on Calvin simulation environment, and the effectiveness of the proposed method is not verified in real-world robot configuration.
2. GFlops sometimes do not reflect the actual operational efficiency well. Reporting the FPS of the model's operation would be better.
3. Does the introduction of the early-exit paradigm weaken the large language model's ability to understand and follow language instruction? This question can be answered by conducting an experiment in the ABCD->D (Enriched) setting, similar to RoboFlamingo, using GPT-4 enriched instructions.

**Questions:**

1. DeeR is built upon the RoboFlamingo++, and all the architecture and hyperparameters are kept consistent with RoboFlamingo++, except for our proposed dynamic early-exit paradigm. Why does DeeR perform better on Calvin than RoboFlamingo++? Why can the early-exit paradigm, designed to improve inference efficiency, also enhance accuracy?
2. The experimental configuration of GR-1 is almost consistent with this paper, except for the extra proprioceptive information employed in GR-1. Why not to implement a "RGB+Proprio" version of DeeR to enable totally fair comparison with GR-1 and observe whether its performance can exceed GR-1.

**Limitations:**

The authors discuss the potential negative societal impacts of their work.

---

> ### Author Rebuttal · Authors · 2024-08-07
>
> Thank you for your time and thoughtful review. Please see the response below.
>
> **W1/W2**:  `the effectiveness of the proposed method is not verified in real-world robot configuration`; `Reporting the FPS of the model's operation`
>
> Thanks. We admit that the submitted version of manuscript limits in the simulated CALVIN benchmark. We are working hard to extend DeeR to real-world robotics tasks. Specifically, we are integrating our early exit paradigm, DeeR, atop the recent framework OpenVLA [3] rather than RoboFlamingo, considering OpenVLA is an open-sourced VLM-based robotic policy originally designed for real-world manipulation. We are currently training it across a set of 25 datasets from the Open X-Embodiment Dataset [1] as OpenVLA and Octo [2] does. This training process is time-intensive, and setting up a real evaluation platform using the Franka robot arm involves a significant workload. Due to the tight timeline of this rebuttal, we regret that we cannot provide comprehensive results at this time.
>
> Although the real scenario experiment is still on the way, we have preliminary results regarding real operational efficiency. We evaluated both RoboFlamingo and DeeR on the same Nvidia V100 GPU.  As the below table shows, **DeeR achieves a 68.1% reduction in LLM inference time compared to RoboFlamingo++, aligning with the theoretical 80.7% reduction in FLOPS.** This assessment was conducted without code optimization of the early-exit implementation. We anticipate that future optimizations will further enhance DeeR’s actual operational efficiency, closely approaching the FLOPS reduction. We will add these results to the revision.
>
> |ABCD->D|Avg Len|Avg LLM GLOPS|Avg LLM inference time|
> |-|-|-|-|
> |RoboFlamingo++|4.07|31.2|55ms|
> |DeeR|4.08|6.0|17.5ms|
>
> Further, we wish to highlight that the **CALVIN benchmark are developed for capturing many challenges present in real-world settings.** It encompasses a wide array of robotic manipulation tasks and features over 400 crowd-sourced natural language instructions, posing significant challenges to SOTA algorithms due to its realistic complexity.
>
> We believe these points collectively demonstrate DeeR's substantial potential for real-world applications.
>
> ---
>
> **W3**: `Does early-exit weaken the large language model's ability to understand and follow language instruction? conducting an experiment in the instruction-enriched setting`
>
> Thank you for your valuable question. In our model, we maintain the LLM parameters fixed and only fine-tune the vision perceiver and vision-language fusion components, following the training paradigm of Flamingo and RoboFlamingo. Therefore, the overall ability of **the whole DeeR model** to follow language instructions should be comparable to that of RoboFlamingo.
>
> However, a potential concern is that since the early-exit strategy often utilizes fewer LLM layers than the full model, the language handling capacity of a shallower LLM configuration might be inadequate when dealing with complex (enriched) instructions. To address this, we can adjust the computational budget to use deeper layers and make DeeR have similar performance with RoboFlamingo++.  As the following results show, **while enriched instructions increase the inference computation cost of DeeR, it still reduces the overall computational cost by ~2.7x compared to RoboFlamingo++ without losing performance**.
>
> |avg len (GFLOPS)|D->D|D->D (Enriched)|
>  |-|-|-|
> |RoboFlamingo++|2.71 (31.2)|1.53 (31.2)|
> |DeeR|2.71 (5.4)|1.53 (11.7)|
>
> ---
>
> **Q1**: `Why does DeeR perform better on Calvin than RoboFlamingo++`
>
> Thanks for your question. DeeR builds upon RoboFlamingo++ and maintains consistent architecture and hyperparameters. The key distinction during training is DeeR's use of the loss function $\mathcal{L}^*$, which randomly samples features over possible exits at each timestep to better mimic dynamic inference scenarios, and an auxiliary loss $\mathcal{L}_\text{aux}$ that optimizes features at intermediate exits (detailed in section 3.3). To further analyze the source of performance gain, we conducted an ablation study below.
>
> It's important to note that variants without $\mathcal{L}^*$ do not support dynamic exit. Therefore, we evaluated all results at the fixed last exit to assess the upper bound performance of the trained model. While these two losses are initially designed to enable effective early inference, we observe that they also marginally enhance overall performance, as shown in the table below. We conjecture that this improvement may be due to the introduction of explicit supervision at shallow network layers, making the training process more effective (similar observations have been made in previous studies [4]). We will make the ablation clearer in the revision.
>
> ||$$\mathcal{L}^*$$|$$\mathcal{L}_\text{aux}$$|Best Len (D->D)|
> |-|-|-|-|
> |DeeR|✔|✔|2.84|
> ||✔||2.80|
> |||✔|2.74|
> |Flamingo++|||2.71|
>
> ---
>
> **Q2**: `Can "RGB+Proprio" version of DeeR exceed GR-1?`
>
> Thanks for your valuable suggestion to enhance DeeR's performance. **Since DeeR was initially designed to improve the inference efficiency of MLLM policies, it was built on the MLLM-based framework RoboFlamingo for a fair baseline comparison without proprioceptive inputs.** We believe the potential of enhancing DeeR with proprioceptive inputs to potentially outperform GR-1. However, due to the limited scope of this paper and the extensive time required to train the "RGB+Proprio" version, we are happy to explore it in future.
>
> [1] Padalkar, Abhishek, et al. "Open x-embodiment: Robotic learning datasets and rt-x models." arXiv preprint arXiv:2310.08864 (2023).
>
> [2] Team, Octo Model, et al. "Octo: An open-source generalist robot policy." arXiv preprint arXiv:2405.12213 (2024).
>
> [3] Kim, Moo Jin, et al. "OpenVLA: An Open-Source Vision-Language-Action Model." arXiv preprint arXiv:2406.09246 (2024).
>
> [4] Zhao, Hengshuang, et al. "Pyramid scene parsing network." CVPR. 2017.

---

> > ### Author Response · Authors · 2024-08-13
> >
> > Thank you for reviewing our paper and your insightful feedback. We are truly honored by your support in accepting it and by your decision to raise your score.

---

### Decision · Program_Chairs · 2024-09-25

**Decision:**

Accept (poster)

**Comment:**

The paper presents a framework called Dynamic Early-Exit for Robotic MLLM (DeeR) aimed at improving the computational efficiency of Multimodal Large Language Models (MLLMs) used in robotic applications. The DeeR framework tries to address the challenge of deploying MLLMs on robots with limited computational resources by dynamically adjusting the size of the activated MLLM based on the complexity of the task.
Authors have satisfactorily addressed most of the concerns raised by the reviewers. Overall a good paper.